# D2FLS-Net: Dual-Stage DEM-guided Fusion Transformer for landslide segmentation

Chengwei Zhao[1][☯], Long Li[1,2,3][☯], Yubo Wang[1], Xuqing Li[1]*, Chong Xu[4], Yubin Song[5], Dongsheng Ren[6], Cheng Xiao[5]

1 School of Remote Sensing and Information Engineering, North China Institute of Aerospace Engineering, Langfang, China, 2 State Key Laboratory of Ecological Safety and Sustainable Development in Arid Lands, Xinjiang Institute of Ecology and Geography, Chinese Academy of Sciences, Urumqi, China, 3 University of Chinese Academy of Sciences, Beijing, China, 4 National Institute of Natural Hazards, Ministry of Emergency Management of China, Beijing, China, 5 School of Electronic and Control Engineering, North China Institute of Aerospace Engineering, Langfang, China, 6 Hebei Provincial Institute of Geological Surveying and Mapping (Spatial Information Technology Application Research Center of Hebei Provincial Bureau of Geology and Mineral Exploration and Development), Langfang, China

☯ These authors contributed equally to this work.
* meililixuqing@163.com

## Abstract

Landslide segmentation from remote sensing imagery is crucial for rapid disaster assessment and risk mitigation. Owing to the pronounced heterogeneity of landslide scales and the subtle visual contrast between some landslide bodies and their background, this task remains highly challenging. Although Transformers surpass convolutional neural networks in modeling long-range contextual dependencies, channel-level or feature-level fusion strategies provide only intermittent terrain cues, leading models to underutilize digital elevation model (DEM) information and to lack fine-grained adaptability to terrain variability. To address this, We propose a Swin-Transformer–based framework, Dual-Stage DEM-guided Fusion Transformer for landslide segmentation (D2FLS-Net), which embeds terrain features via two modules: (1) The Dual-Stage DEM-Guided Fusion (DSDF) module that injects DEM cues twice, where the early stage emphasizes DEM related discontinuities before feature extraction, and the late stage coordinates high-level RGB and DEM semantics through a cross-attention mechanism. (2) The Terrain-aware Pixel-wise Adaptive Context Enhancement (T-PACE) module that optimizes intermediate features using a DEM-gated, pixel-adaptive hybrid of multi-dilation atrous convolutions, enabling broader context aggregation within homogeneous landslide interiors and more precise discrimination at boundaries. We evaluate D2FLS-Net on the Bijie and Landslide4Sense 2022 datasets. On Bijie, the mean Intersection over Union (mIoU) reaches 88.77%, Recall 95.27%, and Precision 94.60%, exceeding the best competing model SegFormer by 7.96%, 7.90%, and 4.05%, respectively. On Landslide4Sense2022, mIoU 72.86%, Recall 82.55%, and Precision 93.30%, surpassing SegFormer by 7.06%, 6.56%, and 5.02%, respectively. Ablation studies indicate that DSDF primarily

**Data availability statement:** All relevant data for this study are publicly available from the Zenodo repository (https://doi.org/10.5281/zenodo.17599856).

**Funding:** This work was supported by the Hebei Provincial Central Government Guide Local Science and Technology Development Fund Project for Free Exploration Basic Research (246Z5402G), Key Project of Hebei Provincial Department of Education (ZD2022089), Doctoral Fund Project of North China Institute of Aerospace Engineering (BKY-2023-03).The funder had no role in study design, data collection and analysis, decision to publish, or preparation of the manuscript.

**Competing interests:** The authors have declared that no competing interests exist.

reduces missed detections of landslide traces, whereas T-PACE refines pixel level context selection. Injecting DEM at the Swin-1 and Swin-4 stages consistently outperforms other stage combinations. In summary, the model shows good detection performance and is suitable for fusing DEM and remote sensing imagery for landslide recognition.

## 1. Introduction

Landslides are gravity-driven mass movements of soil or rock on slopes. They are typically triggered by river undercutting, groundwater activity, earthquakes, or anthropogenic slope excavation. The resulting movement proceeds downslope, either coherently or in a disaggregated manner, along pre-existing or newly formed planes or zones of weakness [1]. According to statistics released by the National Bureau of Statistics of China (https://data.stats.gov.cn/), more than 27,000 geological disasters occurred between 2018 and 2023, over half of which were landslides. Globally, landslides average in excess of 20,000 events per year. They cause roughly 1,000 fatalities, affect more than 900,000 people, and lead to direct economic losses on the order of CNY 20–60 billion [2,3]. Rapid and precise landslide identification is therefore essential for disaster assessment, risk reduction and post-event recovery planning [4,5].

To identify landslides, traditional field surveys, although accurate, are labor-intensive and impractical for large or inaccessible areas. In contrast, remote sensing offers synoptic, repeatable, and cost-effective coverage, enabling efficient mapping of landslide inventories [6,7]. With the advent of high-resolution imagery, machine learning and deep learning have been increasingly adopted to automate landslide segmentation [8,9]. Convolutional neural networks (CNNs), in particular, have achieved success in various remote-sensing segmentation tasks and have been widely explored for landslide mapping [10–13]. Yang et al. [14] compared U-Net, DeepLabv3+ and PSPNet to identify optimal configurations for landslide detection. Building on this, Gao et al. [15] enhanced DeepLabv3+ with an MCC loss function to address class imbalance in remote-sensing scenes, improving detection accuracy. Li [16] further refined DeepLabv3+ with an Atrous Spatial Pyramid Convolution (ASPC) block, which reduce model complexity while strengthening feature extraction and segmentation of landslide regions. Sreelakshmi and Vinod Chandra [17] employed an enhanced U-Net with super-resolution inputs and a visual-saliency–oriented attention mechanism to further improve segmentation. These studies respectively addressed class imbalance in landslide segmentation, reduced model complexity, and introduced attention mechanisms to improve landslide segmentation accuracy. Nevertheless, approaches based on CNNs still face inherent limitations when applied to landslide segmentation. Landslides show substantial variation in scale and morphology, from meter to tens of meters up to kilometer scale hazardous events. Pixels at different positions within a landslide body require different levels of contrast with the background. However, fixed receptive fields in CNNs often struggle to balance broad

context modeling with precise boundary delineation. For example, in mountainous areas with vegetation cover, a landslide body that appears similar to bare soil shows a clear visual contrast with the vegetated background. When the receptive field is too small, critical background information may be missed, which is unfavorable for determining that the pixel at that location belongs to a landslide. When the receptive field is too large, fine details may be overlooked, preventing a precise depiction of the landslide boundary [18–21].

Transformers have recently emerged as a promising alternative. By leveraging self-attention, they capture long-range dependencies and global context, making them attractive for segmenting complex landslides [22]. For example, Tang et al. [23] focused on the automatic detection of coseismic landslides and employed the SegFormer model for this task—this model leverages a Transformer encoder to extract multi-scale features from high-resolution remote sensing imagery. Their experimental results showed that the SegFormer model achieved a mIoU of over 75% even in complex terrain, which not only realized effective automatic detection of coseismic landslides but also fully demonstrated the application potential of the Transformer architecture in landslide detection. Zheng et al. [24] proposed SETR, which employs transformers as encoders to capture global contextual information in each layer and combines it with a simple decoder to construct a semantic segmentation network. Liu et al. [25] proposed a novel Vision Transformer model that incorporates a layered architecture and windowed self-attention mechanisms, enhancing modeling flexibility across different scales through multi-scale windows. These researchers used methods based on Transformers to effectively integrate global information and achieve landslide semantic segmentation. However, these studies remain confined to RGB three-band optical inputs. Although landslides in forested areas are generally less frequent, regions such as southwestern China, Nepal, and Japan lie near active fault zones and experience extreme rainfall and other climatic drivers. Together with ecological fragmentation, these factors mean that landslides still occur frequently despite vegetation cover. After a landslide, the surface may be revegetated over time, or weathering and erosion may reduce spectral contrast, so the landslide body can become visually indistinguishable from the background in optical imagery. Therefore, the landslide-recognition community has begun to focus on multi-source data fusion, and the integration of auxiliary geospatial data has been shown to be beneficial for detecting these cases. Liu et al. [26] reported significant improvements in detecting "old, visually blurred" landslides by incorporating DEM-derived terrain features into a segmentation network. Wu et al. [27] likewise demonstrated that combining high-resolution imagery with DEM data and employing attention-enhanced CNNs outperformed imagery-only approaches. However, Li et al. [28] found that simple early fusion like stacking DEM as an additional channel often fails to fully exploit topographic cues and may even introduce noise, which can cause the Hughes phenomenon and reduce the recognition performance for landslides whose bodies show clear visual differences from the surrounding background. Some more advanced schemes adopt late fusion or parallel fusion [29], but during feature extraction these approaches can lead to loss of DEM information, thereby limiting the model's discriminative capability in complex terrain where the visual differences between landslide bodies and background are small.

To address the above issues, this study proposes a D2FLS-Net, and evaluates its performance on Bijie and Landslide-4Sense2022 datasets. The main contributions are as follows:

(1) We construct a Dual-Stage DEM-Guided Fusion (DSDF) module, which in the early stage guides and assists optical semantic segmentation for landslide recognition in the form of weights, and in the later stage enables interaction with optical features at the high level semantic feature stage, ultimately achieving effective fusion of optical data and DEM data, overcoming the limitations of one time fusion and ensuring persistent topographic guidance throughout the process from low level edge features to high level semantic features.

(2) We construct a Terrain-aware Pixel-wise Adaptive Context Enhancement (T-PACE) module, which addresses the limited receptive-field range of dilated (atrous) convolutions; through DEM-gated multi-dilation convolutions, the model is endowed with dynamic, location-specific receptive fields, enabling fine boundary preservation.

(3) On the Bijie Landslides and Landslide4Sense2022 datasets, our model achieves superior landslide recognition performance compared with SAM, SegFormer, U-Net, HRNet, and ResNet-50.

## 2. Method

We propose Dual-Stage DEM-guided Fusion Transformer for Landslide Segmentation (D2FLS-Net), a binary semantic segmentation network built on a Swin-Transformer backbone with Swin Stages 1–4. The DSDF module injects topographic guidance at two levels. In the shallow stage, early DEM guidance emphasizes slope-related discontinuities prior to RGB feature extraction. In the deep stage, a high-level cross-attention aligns deep RGB and DEM semantics. The T-PACE module refines mid-level features via a DEM-gated, pixel-adaptive mixture of parallel dilated convolution branches with different dilation rates (multi-dilation atrous branches), enabling per-pixel selection of the most suitable receptive field. Together, DSDF and T-PACE enhance boundary localization—i.e., precise edge delineation for vegetated and ancient landslides, while preserving robustness on new landslides.

### 2.1. Architecture for transformer network based on dual-Stage DEM guidance and fusion optimization

D2FLS-Net consists of four parts. (i) a Swin-Transformer backbone that produces multi-scale image features at Swin stages1-4; (ii) the DSDF module, which applies early DEM guidance at Swin stage-1 and high-level cross-attention at Swin stage-4; (iii) the T-PACE module mounted at Swin stage-3; and (iv) a lightweight Feature Pyramid Network (FPN) decoder head, a standard module for multi-scale feature fusion, to generate the final two-channel mask. RGB image $X \in \mathbb{R}^{B \times 3 \times H \times W}$, DEM $D \in \mathbb{R}^{B \times 1 \times H \times W}$. The Swin backbone outputs $\{F_1^I, F_2^I, F_3^I, F_4^I\}$ with spatial resolutions approximately $(H/4, H/8, H/16, H/32)$ and channel dimensions (96, 192, 384, 768) for the Swin-Transformer [30]. The overall architecture is shown in Fig 1. The two DSDF stages are detailed in Fig 2 and T-PACE in Fig 3.

The Swin-Transformer is a hierarchical Vision Transformer that employs shifted-window self-attention. Let $\varphi_s(\cdot)$ denote the stack of Swin block at stage $s \in \{1, 2, 3, 4\}$. With a patch embedding of stride 4, the backbone computes stage-wise features: $F_1^I = \varphi_1(\hat{X})$, $F_2^I = \varphi_2(F1^I)$, $F_3^I = \varphi_3(F_2^I)$, $F_4^I = \varphi_4(F_3^I)$, where $\hat{X}$ denotes the RGB input optionally preconditioned by early DEM guidance. For conciseness, we treat the $\varphi_s$ stacks as a black box and focus on how DSDF and T-PACE interact with $F_1^I$, $F_3^I$, and $F_4^I$. The DEM branch encodes D using a tiny ViT-style transformer encoder (two layers, width 128, heads 4; ViT) following a 4×4 patch embedding with stride 4. The resulting feature map Z is resized and linearly projected 1×1 to obtain DEM-aligned features $F_3^D \in \mathbb{R}^{B \times 384 \times H/16 \times W/16}$ and $F_4^D \in \mathbb{R}^{B \times 768 \times H/32 \times W/32}$.

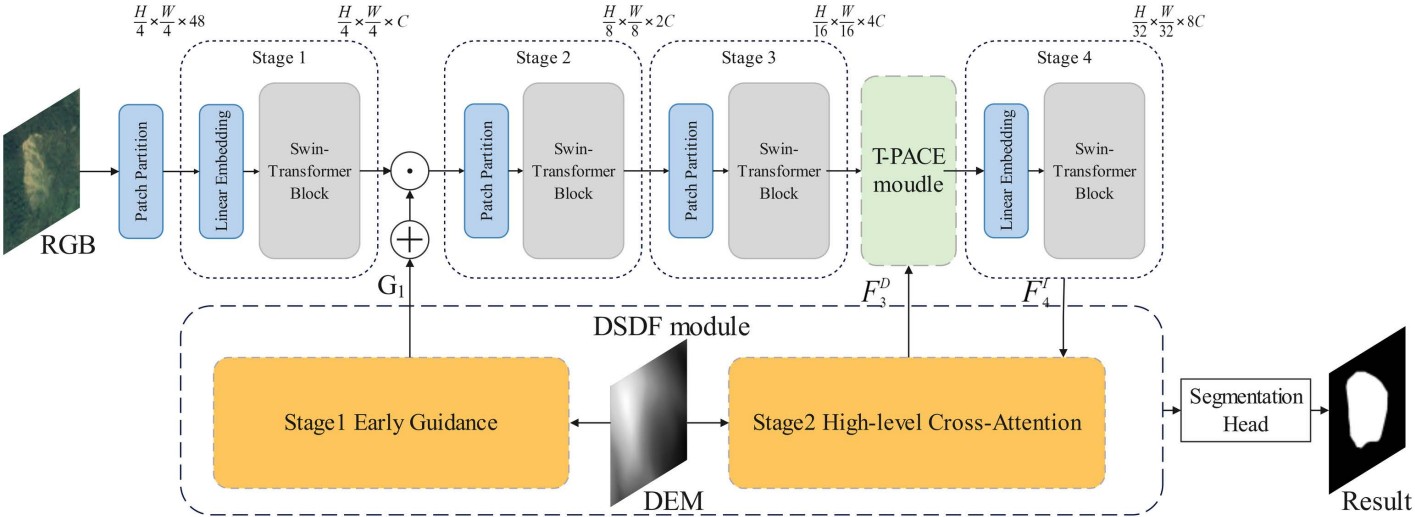

**Fig 1. Architecture for transformer network based on dual-stage DEM guidance and fusion optimization.**

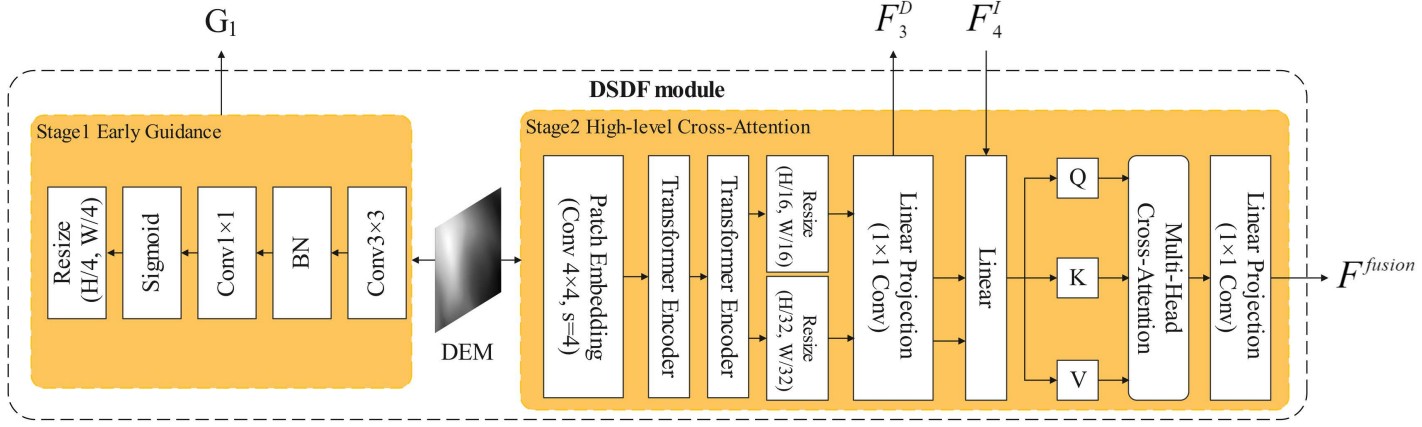

**Fig 2. Dual-Stage DEM-Guided Fusion module.**

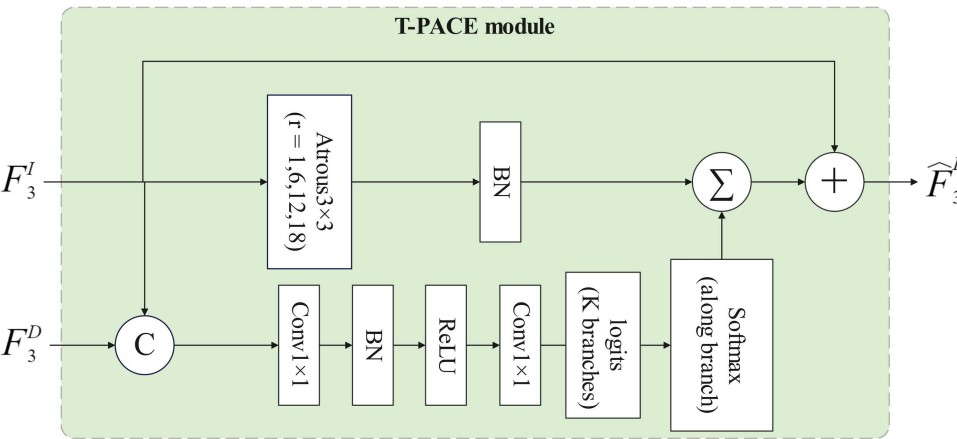

**Fig 3. Terrain-aware Pixel-wise Adaptive Context Enhancement module.**

We optimize the network using the pixel-wise cross-entropy loss on two classes (landslide and background). Given prediction $\hat{y}_{p,c}$ and one-hot target $y_{p,c}$ at pixel $p$ for class $c \in \{0, 1\}$, the loss is calculated as Equation (1).

$$L_{CE} = -(1/N) * \sum_{p=1}^{N} \sum_{c\in\{0,1\}} y_{p,c}\log(\hat{y}_{p,c})$$

(1)

For highly imbalanced scenes, we optionally combine LCE with Dice loss to stabilize boundary learning, as shown in Equation (2) and Equation (3).

$$L_{Dice} = 1 - (2\sum_{p} y_{p,1}\hat{y}_{p,1} + \varepsilon)/(\sum_{p} y_{p,1} + \sum_{p} \hat{y}_{p,1} + \varepsilon)$$

(2)

$$L = L_{CE} + \lambda L_{Dice}$$

(3)

Here, $\varepsilon$ is a small constant for numerical stability, $\lambda \geq 0$ is a weighting factor. Class 1 denotes the landslide class.

We adopt a lightweight FPN head a standard multi-scale feature fusion module to generate the final two-channel mask. First, $F_4^{\text{fusion}}$ is bilinearly upsampled to the Swin stage-3 resolution and concatenated along the channel dimension with the T-PACE output $\hat{F}_3^I$. Next, a 1×1 convolution reduces the channel dimension, followed by two successive transposed convolutions (kernel $2 \times 2$, stride 2) to reach H/4. Finally, a final 1×1 classifier predicts two logits (the raw scores before converting to class probabilities) which are bilinearly upsampled to H×W.

## 2.2. Dual-stage DEM-guided fusion module

From a geomechanical perspective, the causes and expressions of landslides operate at different feature levels. At shallow levels, local slope and aspect modulate the spectral textures of bare soil, colluvium, and vegetation edges. In this regime, the DEM can immediately suppress implausible feature channels (e.g., a steep-slope preference) and highlight likely scarps or deposits. At higher levels, the presence, shape, and continuity of landslide bodies must be reasoned jointly across modalities. For example, elongated planar deposits visible in RGB should be linked to persistent topographic signatures in the DEM. Cross-attention between deep tokens is well-suited to this abstraction. This staged design mirrors multimodal fusion patterns recently reported in remote sensing. In particular, feature reconstruction and dual-branch extraction mitigate redundancy across modalities. We extend this idea to segmentation and to the DEM–RGB pair.

Landslides in densely vegetated or eroded terrains often exhibit weak radiometric contrast. The DEM captures terrain morphology (elevation and slope) that complements RGB texture. Fusing DEM features only at the low-level channel stage or at the high-level semantic stage prevents landslides with diverse manifestation from benefiting simultaneously and reliably from DEM-derived topographic cues. We therefore inject DEM twice, with distinct roles: (1) Before feature extraction, a spatial guidance map boosts slope-related regions, thereby enabling early filters to attend to topography-sensitive pixels. (2) At the highest semantic layer, cross-attention aligns modality-specific abstractions, allowing RGB queries to retrieve DEM contexts consistent with large-scale geomorphology. This two-stage design reduces both early under-attention to subtle landslides and late semantic disagreements between modalities.

### 2.2.1. Early DEM guidance.

Early DEM guidance does not fully fuse DEM in a way that would disturb landslides with clear visual differences. Instead, it acts in a weighted form to assist the optical data in the semantic segmentation of landslides. This multiplicative gain retains radiometric cues while emphasizing terrain-salient pixels, improving downstream feature extraction for low-contrast landslides.

Given DEM $D$, we compute a guidance map $G$ by a shallow CNN and modulate Swin stage-1 features. Let $H \times W$ denote the full resolution and $(\frac{H}{4}, \frac{W}{4})$ the stage-1 stride. We first produce $G \in [0, 1]^{B \times 1 \times H \times W}$, calculated as Equation (4).

$$G = \sigma(Conv_{1 \times 1}(BN(Conv_{3 \times 3}(D))))$$

(4)

Here, $Conv_{k \times k}$ denotes a k×k convolution. BN is batch normalization. sigmoid(·) maps to [0,1]. D has shape $(B, 1, H, W)$. G matches $(B, 1, H, W)$.

We resize $G$ to stage-1 resolution, $G1 = Resize(G; H/4, W/4)$, and apply pixel-wise gain on the RGB stage-1 tensor, calculated as Equation (5)

$$\hat{F}1^I = (1 + G1) \odot F1^I$$

(5)

Here, Resize(·; h,w) bilinearly resizes its input to the spatial size $(h, w)$. The operator $\odot$ denotes element-wise (Hadamard) product with broadcasting along the channel dimension. The feature map $F_1^I \in \mathbb{R}^{B \times 96 \times H/4 \times W/4}$. Adding 1 is a gain, so when $G1 = 0$ there is no suppression.

In plain terms, the early DEM map boosts slope-salient areas before feature extraction, so the backbone pays extra attention to places where landslides are physically plausible.

**2.2.2. High-level cross-attention.** High-level cross-attention fuses DEM and optical data once again at the high level semantic feature stage. It enables interaction between the two features to address the missed detection problems for landslides with blurred boundaries or vegetation occlusion. At this point, the DEM features have been sparsely extracted and abstracted to a certain level, and likewise will not interfere with the recognition of landslides that show clear visual differences.

The DEM branch encodes $D$ via a patch embedding ($4 \times 4$, stride 4) and a tiny ViT-style transformer encoder (two layers, width 128, 4 heads). The encoded feature map $Z$ is resized and projected ($1 \times 1$) to align with Swin stage-3/4, calculated as shown in Equation (6) and Equation (7).

$$F_3^D = Conv_{1\times1}^{E\to384}(Resize(Z; H/16, W/16)) \tag{6}$$

$$F_4^D = Conv_{1\times1}^{E\to768}(Resize(Z; H/32, W/32)) \tag{7}$$

where E is the DEM encoder width (default 128). $Conv_{1\times1}^{E\to384}$ is a $1 \times 1$ convolution mapping $E$ channels to $C$. $F_3^D$ and $F_4^D$ align to Swin stage-3/4 sizes, respectively.

At Swin stage-4, we derive $Q$ from RGB and $K$, $V$ from DEM and perform multi-head cross-attention (MHCA), calculated as shown in Equation (8) and Equation (9). Cross-attention here can be read as letting RGB 'ask questions' and the DEM 'provide context', so high-level RGB features retrieve terrain cues that are consistent with large-scale geomorphology.

$$Q = WQ * F_4^I, K = WK * F_4^D, V = WV * F_4^D \tag{8}$$

$$Attn(Q, K, V) = softmax((QK^T)/\sqrt{d})V \tag{9}$$

where WQ, WK, WV are $1 \times 1$ convolution (or linear) projections applied per spatial location to map channels to the attention head dimension. "*" denotes such linear projection, not spatial convolution.

The attention output is projected back to the stage-4 channel size ($1 \times 1$), yielding the fused high-level feature, calculated as shown in Equation (10).

$$F^{fusion} = WO * Attn(Q, K, V) \tag{10}$$

where WO is the output $1 \times 1$ projection (a linear layer) mapping concatenated heads back to C = 768 at Swin stage-4.

This design lets RGB queries gather terrain-consistent support from DEM at large receptive fields while leaving mid-level geometric refinements to T-PACE.

## 2.3. Terrain-aware pixel-wise adaptive context enhancement

In forests or humid mountains, old landslides are often re-vegetated, and RGB boundaries become indistinct. Interior pixels of such bodies benefit from broad contextual support to avoid fragmenting the mask, whereas boundary pixels require detail-preserving fields of view. A purely RGB-driven selector can fail in heavily textured backgrounds. Incorporating terrain cues improves the per-pixel decision on how much context to aggregate. Unlike a fixed dilation set, T-PACE conditions both channel weights and dilation emphasis on DEM features, providing a physically meaningful prior that stabilizes adaptation across scenes.

 

T-PACE enhances $F_3^I$ at Swin stage-3 by aggregating multi-dilation contexts with DEM-aware pixel-wise gating. Let dilations $r_k \in \{1, 6, 12, 18\}$ and $K$ the number of branches. For each branch, as shown in Equation (11).

$$Z_k = BN(Conv_{3\times3}^{r_k}(F_3^I)), k = 1..K \tag{11}$$

where $Conv_{3\times3}^{r_k}$ is a $3 \times 3$ atrous convolution with dilation $r_k$; BN is batch normalization; $Z_k$ shares shape (B,384,H/16,W/16). K = 4 by default.

We form a gating tensor by concatenating RGB and DEM features $U$, calculated as shown in Equation (12). Passing through $1 \times 1$ convolutions to produce per-pixel logits over branches, followed by softmax [31] along the branch dimension, calculated as shown in Equation (13) and Equation (14).

$$U = [F_3^I || F_3^D] \tag{12}$$

$$L = W2 * BN(\sigma(W1 * U)) \tag{13}$$

$$\alpha = softmax_{\text{branch}}(L) \tag{14}$$

Here, "||" denotes channel-wise concatenation. W1 and W2 are $1 \times 1$ convolutions. ReLU denotes the rectified linear unit. The coefficients $\alpha \in \mathbb{R}^{B \times K \times H/16 \times W/16}$ are normalized per pixel, i.e., $\sum_{k=1}^{K} \alpha_k(i,j)$ for all $(i,j)$.

Pixel-adaptive aggregation and residual refinement then yield the enhanced stage-3 representation, calculated as shown in Equation (15)–(17).

$$Y = \sum_{k=1}^{K} \alpha_k \odot Z_k \tag{15}$$

$$P = BN(Conv_{1\times1}(Y)) \tag{16}$$

$$\hat{F}_3^I = P + F_3^I \tag{17}$$

where $\Sigma$ is summation over branches; $\odot$ is element-wise product broadcasting $\alpha_k$ over channels; $Conv_{1\times1}$ is a $1 \times 1$ projection to C = 384; "+" is residual element-wise addition.

Compared with fixed ASPP modules, T-PACE learns a per-pixel mixture over receptive fields conditioned on DEM, enabling selective emphasis on boundaries versus interior regions.

## 3. Experiments

### 3.1. Data description

We evaluate on two public benchmarks. (1) The Bijie landslide dataset covers Bijie City in northwestern Guizhou, China (~26,853 km²), a transition zone from the Tibetan Plateau to eastern hills with elevations ranging from ~457–2900 m. Due to complex lithology, steep slopes, annual precipitation of ~849–1399 mm, and fragile ecosystems, the area experiences frequent failures dominated by rockfall and rock slide, with occasional debris flows. The dataset comprises satellite RGB imagery, expert-annotated polygon masks, and a DEM layer, totaling 770 landslide samples and 2003 non-landslide samples. RGB images were collected by TripleSat between May and August 2018 at 0.8 m GSD; the DEM has ~2 m vertical

accuracy. Landslide outlines were digitized in ArcGIS using both RGB and DEM cues. (2) The Landslide4Sense 2022 benchmark aggregates multi-sensor scenes from several landslide-affected regions between 2015 and 2021 (Iburi-Tobu, Hokkaido; Kodagu, Karnataka; Rasuwa, Bagmati; and western Taitung). It provides 14 layers resampled to 10 m: Sentinel-2 multispectral bands (B1–B12), a DEM, and slope derived from ALOS PALSAR. Pixel-wise binary labels (landslide/non-landslide) are available. Landslide4Sense2022 has its training subset drawn from four typical events induced by different triggering factors and distributed across multiple regions, and the organizers have withheld higher resolution location details to avoid external comparison, this design inherently emphasizes the evaluation of generalization across regions and across triggering factors. Compared with the Bijie dataset that provides fine annotation for individual landslide objects, Landslide4Sense2022 is more engineering oriented and focuses on scene level evaluation, where a single training sample often contains contiguous landslide areas and complex backgrounds, which can better examine the generalization capability of methods. These two datasets jointly probe our method's behavior in ultra-high-resolution mountainous settings and in multi-modal medium-resolution scenarios. As shown in Fig 4, it displays the study area corresponding to the dataset we used.

### 3.2. Experimental setup

#### 3.2.1. Evaluation metric.
This study evaluates pixel-level binary semantic segmentation into landslide and background classes. Let $y(p) \in \{0, 1\}$ denote the ground-truth label for pixel $p$. We set $y(p) = 1$ for landslide and $y(p) = 0$ for background. Let $\hat{y}(p) \in \{0, 1\}$ denote the classified label, obtained by argmax over the model's softmax scores at pixel $p$. For sigmoid

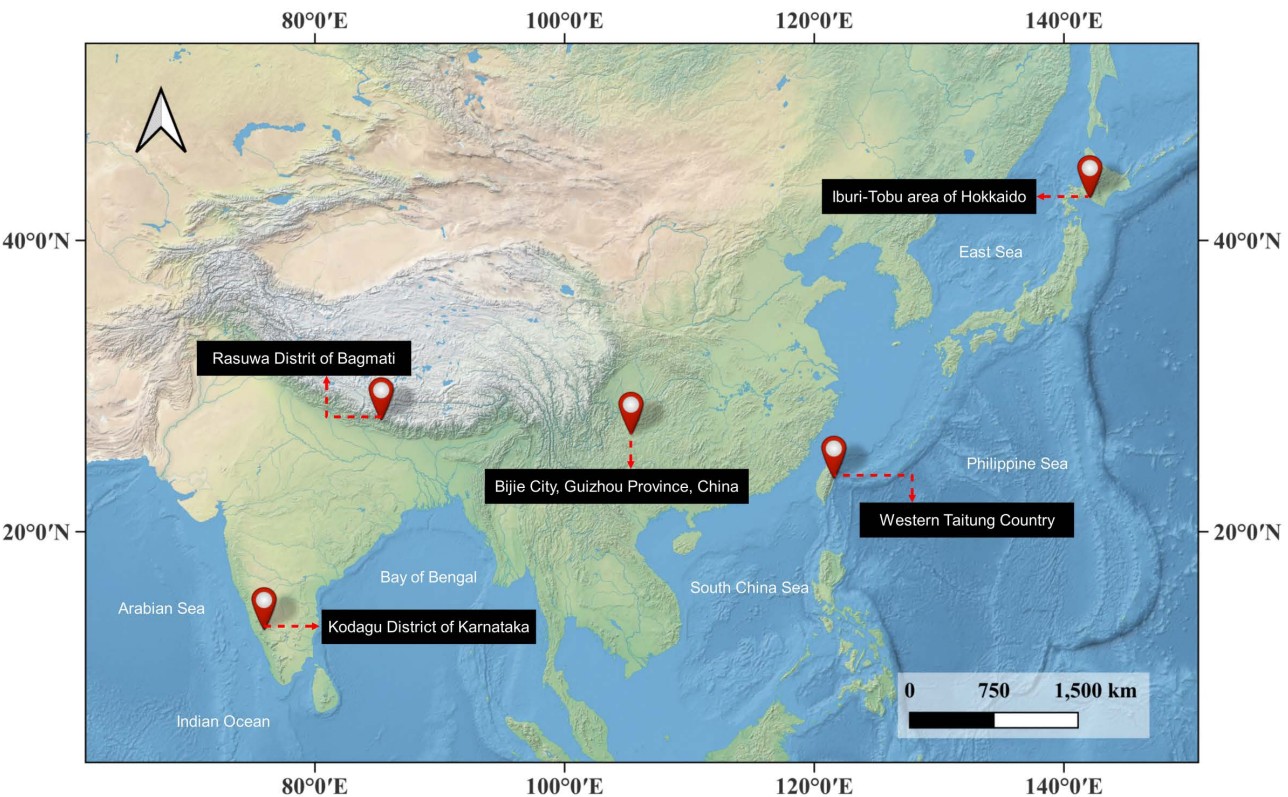

**Fig 4. Study area overview.** (Points of interest are shown over an open basemap. Basemap: Natural Earth II, public domain. Made with Natural Earth. Figure © Authors, CC BY 4.0.).

outputs, we use a threshold of 0.5. In the test set, we count four pixel-level outcomes. True positive (TP) is a pixel predicted as landslide whose ground-truth label is landslide. False positive (FP) is predicted as landslide but its ground-truth label is background. True negative (TN) is predicted as background and its ground-truth label is background. False negative (FN) is predicted as background while its ground-truth label is landslide. We denote these counts by TP, FP, TN, and FN, respectively.

We report the Intersection-over-Union (IoU) for each class, the mean IoU (mIoU), defined in Equation (18)–(20). With a small $\varepsilon > 0$ for numerical stability,

$$IoU_{landslide} = TP/(TP + FP + FN + \varepsilon) \tag{18}$$

$$IoU_{background} = TN/(TN + FP + FN + \varepsilon) \tag{19}$$

$$mIoU = (IoU_{landslide} + IoU_{background})/2_{\circ} \tag{20}$$

Precision and Recall are computed as Equation (21) and (22).

$$Precision = TP/(TP + FP + \varepsilon) \tag{21}$$

$$Recall = TP/(TP + FN + \varepsilon) \tag{22}$$

**3.2.2. Experimental settings.** In this study, each dataset was processed independently. For every dataset, the samples were partitioned into training, validation, and test subsets in a 6:2:2 ratio (60%, 20%, 20%).All models were implemented in PyTorch 1.8 with torchvision 0.9 and CUDA 11.1 on Ubuntu 22.04. Unless otherwise stated, the optimizer was AdamW with an initial learning rate of $5 \times 10^{-5}$, weight decay of 0.01, and a mini-batch size of 24. All experiments were executed under an identical software/hardware configuration: a single NVIDIA GeForce RTX 3090 GPU (24 GB) and an Intel® Xeon® Gold 6248R CPU at 3.00 GHz.

## 3.3. Results and analysis

**3.3.1. Ablation studies with backbone scaling.** Table 1 summarizes the backbone configurations of the three D2FLS-Net variants used in our scaling study, namely D2FLS-Net-S, D2FLS-Net-B, and D2FLS-Net-L, including stage depths, the number of heads, and the embedding dimension. We use these variants to isolate the effect of capacity while keeping the fusion design and the training protocol fixed.

Table 2 summarizes a two-factor study. (i) module ablations (Swin backbone only; with T-PACE; with DSDF and full D2FLS-Net), and (ii) backbone capacity (Swin-S, Swin-B, Swin-L). Across all three capacities, the full model, namely Swin with DSDF and T-PACE, yields the best results, and averaging over Swin-S, Swin-B, and Swin-L on Bijie the full model

**Table 1. Parameters of the backbone structures of D2FLS-Net -L, D2FLS-Net -B and D2FLS-Net -S.**

| Backbone | D2FLS-Net -L | D2FLS-Net -B | D2FLS-Net -S |
|---|---|---|---|
| Embed Dim | 192 | 128 | 96 |
| Depths | [2 2 18 2] | [2, 2, 18, 2] | [2, 2, 18, 2] |
| Num Heads | [6, 12, 24, 48] | [4, 8, 16, 32] | [3, 6, 12, 24] |
| Window Size | 7 | 7 | 7 |

**Table 2. Experimental results on the Bijie dataset.**

| Backbone | Swin-block | DSDF | T-PACE | MIoU(%) | Recall(%) | Precision(%) |
|---|---|---|---|---|---|---|
| **Swin-L** | √ | | | 75.45 | 80.70 | 93.36 |
| | √ | | √ | 79.91 | 86.13 | 90.23 |
| | √ | √ | | 86.66 | 92.78 | 93.58 |
| | √ | √ | √ | 88.54 | 94.31 | 93.95 |
| **Swin-B** | √ | | | 75.98 | 81.27 | 93.34 |
| | √ | | √ | 73.14 | 82.35 | 89.9 |
| | √ | √ | | 82.91 | 88.81 | 93.25 |
| | √ | √ | √ | 88.77 | 95.27 | 94.6 |
| **Swin-S** | √ | | | 73.25 | 79.68 | 92.71 |
| | √ | | √ | 79.87 | 86.47 | 90.77 |
| | √ | √ | | 78.35 | 88.34 | 92.1 |
| | √ | √ | √ | 88.02 | 94.55 | 93.86 |

improves mIoU, recall, and precision by +13.6%, +14.2%, and +1.0% over the plain Swin block. Decomposing contributions, DSDF only accounts for most of the recall gain (+9.4% on average) with negligible precision change (−0.16%), whereas T-PACE only provides a smaller gain in mIoU and recall (+2.7% and +4.4%) at the cost of precision (−2.8%). This indicates that dual stage DEM guidance chiefly reduces false negatives on boundary blurred scars, while pixel adaptive context controls boundary decisions. T-PACE in isolation primarily boosts recall with a modest precision trade-off, whereas adding DSDF recovers precision by injecting topographic priors, leading to a balanced improvement on both axes. Notably, the medium capacity variant, Swin B, attains the highest mIoU on Bijie, but the gap among full models with Swin-S, Swin-B, and Swin-L is about 1%, indicating that the two proposed modules rather than the backbone size drive the gains.

Table 3 shows the same ablation results on the medium resolution Landslide4Sense2022 dataset under the same experimental settings as the Bijie dataset. On Landslide4Sense2022, averaging over Swin-S, Swin-B, and Swin-L, the full model improves mIoU, recall, and precision by +5.5%, +6.1%, and +12.6% over the plain Swin baselines. In contrast, partial configurations that add a single module can yield mixed effects. T-PACE only consistently raises precision by +9.5% on average while lowering recall by −3.9%, and DSDF only slightly decreases mIoU by −1.3% with a larger recall drop of −6.9%. We attribute this to the DEM cues being coarser than the target size at 10m resolution. Moreover, an overall

**Table 3. Experimental results on the Landslide4Sense2022 dataset.**

| Backbone | Swin-block | DSDF | T-PACE | MIoU(%) | Recall(%) | Precision(%) |
|---|---|---|---|---|---|---|
| **Swin-L** | √ | | | 65.01 | 72.10 | 84.89 |
| | √ | | √ | 64.99 | 69.75 | 92.88 |
| | √ | √ | | 62.88 | 65.28 | 85.36 |
| | √ | √ | √ | 72.86 | 82.55 | 93.30 |
| **Swin-B** | √ | | | 66.60 | 73.61 | 80.92 |
| | √ | | √ | 66.58 | 70.81 | 91.09 |
| | √ | √ | | 62.93 | 66.10 | 83.05 |
| | √ | √ | √ | 71.13 | 81.87 | 95.51 |
| **Swin-S** | √ | | | 66.42 | 82.03 | 81.37 |
| | √ | | √ | 67.95 | 75.38 | 91.66 |
| | √ | √ | | 68.40 | 75.66 | 84.17 |
| | √ | √ | √ | 70.50 | 81.70 | 96.05 |

inspection of the dataset indicates that most landslides exhibit pronounced visual contrast between the landslide body and the background. Methods with DSDF only tend to over regularize toward terrain priors, yielding more false negatives, whereas T-PACE alone becomes conservative, yielding fewer false positives. The proposed DSDF, together with T-PACE, rebalances this bias variance trade-off through dual stage DEM fusion, in which spatially designed feature interactions between DEM and optical data achieve complementary strengths.

**3.3.2. Ablation studies on the data fusion stages.** To verify the choice of injecting DEM guidance at Swin 1 (early) and Swin 4 (late), we conducted a controlled ablation that enumerates all two stage combinations {(1,2), (1,3), (1,4), (2,3), (2,4), (3,4)} while keeping all other settings identical, namely the backbone, T PACE at Swin 3, the decoder, and the training protocol. As summarized in Table 4, the proposed (1,4) pairing consistently attains the best trade-off among mIoU, recall, and precision on both datasets. On Bijie with 0.8 m ground sampling distance, alternatives that omit the earliest stage, for example (2,4) and (3,4), lose boundary fidelity and reduce recall and mIoU; alternatives that omit the deepest stage, for example (1,2) and (1,3), show weaker semantic alignment and more false alarms, slightly lowering precision. On Landslide4Sense 2022 with 10 m resolution and multi sensor inputs, (1,4) again dominates, whereas (2,4) tends to favor precision at the cost of recall, and (1,3) improves recall but lacks global semantic checks, leading to modest precision drops. These trends support our design intent, early DEM priors guide shallow filters toward slope salient regions, and late cross attention reconciles modality semantics at the most expressive stage. Together, this dual stage scheme yields the most stable and accurate performance across diverse terrain and scale conditions.

**3.3.3. Comparison with representative methods.** To validate the landslide recognition performance of the proposed D2FLS-Net, we selected representative algorithms from both CNN-based and Transformer-based families for quantitative and visualization-based comparisons.

Fig 5 shows eight representative scenes from the Bijie test set, arranged as Image/DEM/Label followed by predictions from D2FLS-Net, SAM [32], SegFormer [33], Unet [34], HRNet [35], and ResNet50 [36]. Rows (a)–(c) depict landslides that have been covered again by vegetation or have been weathered, whose RGB boundaries are scarcely visible. Owing to the designed DSDF module, D2FLS-Net ensures persistent injection of elevation data and effective fusion with RGB data, allowing better consideration of the topographic characteristics of landslides. Therefore, it recovers compact masks with fewer holes and better adherence to the DEM-indicated scar geometry, whereas the other baselines tend to fragment the target. In (d), where vegetation partially re-appears on the landslide body, our method suppresses adjacent roads exhibiting homogeneous optical reflectance and penetrates the confounding vegetation clutter present on the landslide body, yielding a tighter outline. Rows (e)–(h) correspond to visually salient, high-contrast landslides. All methods locate the

**Table 4. Experimental results for different stage combinations.**

| Swin-B | Stage 1 | Stage 2 | Stage 3 | Stage 4 | MIoU(%) | Recall(%) | Precision(%) |
|---|---|---|---|---|---|---|---|
| **Bijie dataset** | √ | √ | | | 85.11 | 93.12 | 92.03 |
| | √ | | √ | | 97.95 | 93.80 | 93.85 |
| | √ | | | √ | 88.77 | 95.27 | 94.6 |
| | | √ | √ | | 84.32 | 92.24 | 91.71 |
| | | √ | | √ | 87.42 | 93.92 | 96.50 |
| | | | √ | √ | 86.19 | 93.17 | 93.47 |
| **Landslide4Sense2022 dataset** | √ | √ | | | 68.80 | 78.81 | 90.53 |
| | √ | | √ | | 70.86 | 80.54 | 92.70 |
| | √ | | | √ | 71.13 | 81.87 | 95.51 |
| | | √ | √ | | 68.22 | 77.95 | 91.97 |
| | | √ | | √ | 70.48 | 79.96 | 93.86 |
| | | | √ | √ | 69.61 | 78.43 | 92.75 |

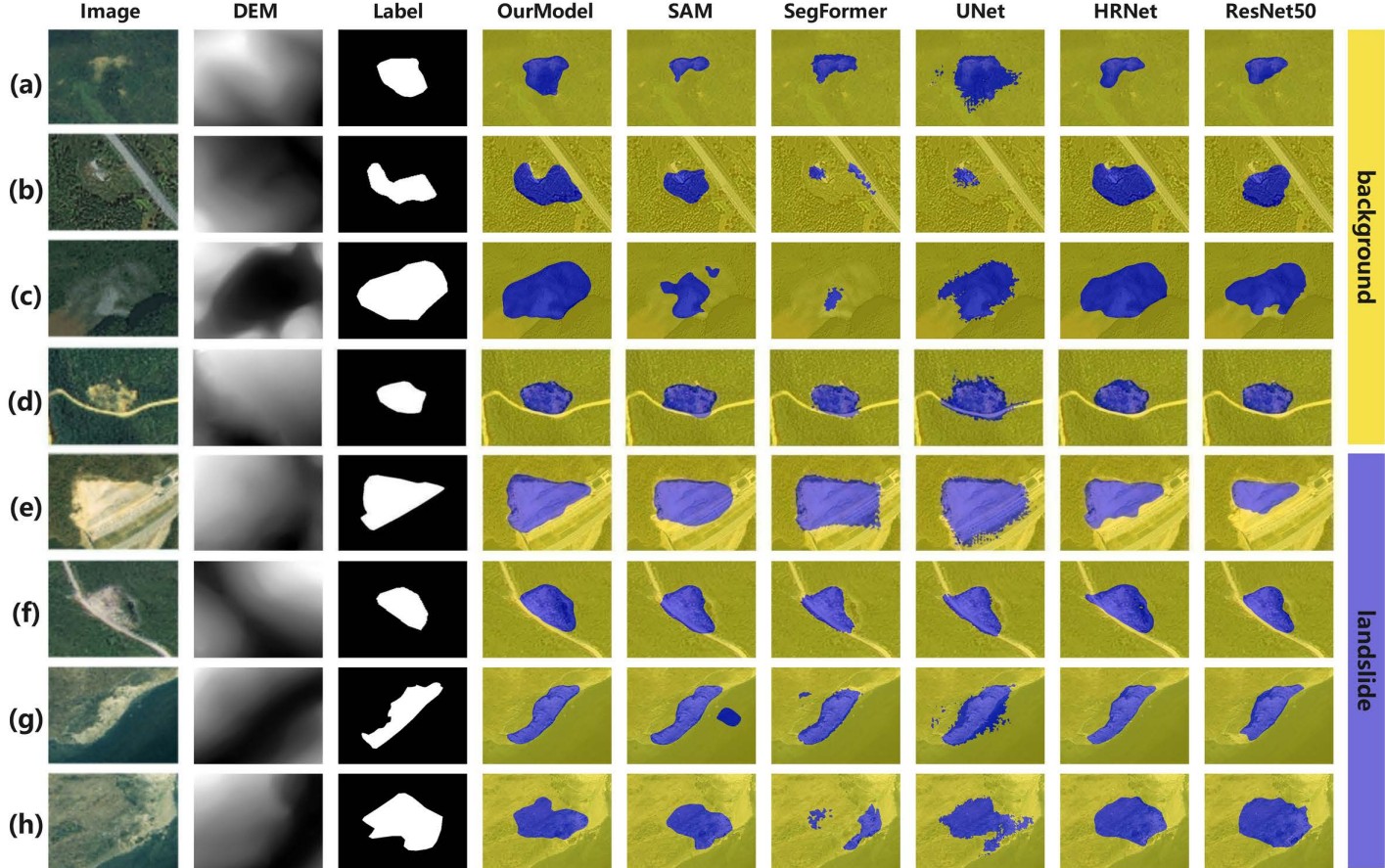

**Fig 5. Visual comparison on the Bijie dataset.** Blue＝landslide, Yellow＝background. (Contains modified Copernicus Sentinel-2 data, processed by the authors. No TripleSat pixels are reproduced; masks were trained/tested using the Bijie dataset. This replacement image is similar but not identical to the original and is for illustrative purposes only. Figure © Authors, CC BY 4.0).

bulk of the target, but D2FLS-Net, owing to its designed T-PACE module, can adjust the necessary field of view according to position, thereby balancing global and local features, and it produces cleaner borders and fewer spurious responses on surrounding bare soil and roads. Overall, the proposed method identifies landslide bodies in their entirety while preserving smooth boundaries, and it adaptively fuses topographic features for boundary delineation. These qualitative trends are consistent with the design goals of DSDF and T-PACE, which together enhance boundary localization under weak radiometric contrast.

Table 5 reports each model's mIoU, recall, and precision on the Bijie landslide dataset. D2FLS-Net achieves the best performance, with mIoU 88.77%, recall 95.27%, and precision 94.60%, surpassing the strongest baseline SegFormer with mIoU 80.81% by 7.96%, and outperforming SAM, U Net, HRNet, and ResNet 50 by 10.49%, 11.79%, 10.60%, and 12.37% in mIoU, respectively. The recall gain is particularly pronounced, at least 7.9% across all baselines, while precision remains the highest among the compared methods. This indicates improved sensitivity to boundary blurred scars, and that DSDF and T PACE are complementary and reduce missed and false detections, resulting in fewer false negatives and fewer false positives. It is unaffected by whether the landslide body and the background exhibit visual contrast, and it achieves higher metrics. By contrast, other methods display more conservative decision making, sacrificing recall and mIoU to attain high precision.

**Table 5. Quantitative comparison on the Bijie dataset.**

| Method | MIoU(%) | Recall(%) | Precision(%) |
|---|---|---|---|
| D2FLS-Net | 88.77 | 95.27 | 94.60 |
| SAM (ViT-B) | 78.28 | 83.83 | 90.68 |
| SegFormer-B2 | 80.81 | 87.37 | 90.55 |
| UNet | 76.98 | 83.48 | 90.34 |
| HRNet | 78.17 | 84.86 | 90.79 |
| ResNet50 | 76.4 | 84.2 | 86.89 |

Fig 6 summarizes eight scenes from Landslide4Sense2022. In this medium resolution and multi sensor setting, landslides are often small relative to the scene and class imbalance is severe. D2FLS-Net preserves small detachments and elongated scars with fewer false alarms, while several baselines either miss small targets, such as SAM and UNet, or confuse bare soil with landslides, as marked by red boxes for SAM, SegFormer and UNet. HRNet produces sharp edges but frequently under-detects scattered fragments, especially in terraced or shadowed areas. Qualitatively, the DEM-aware cross-attention at the top stage helps our model latch onto slope-consistent structures, and T-PACE limits context leakage from adjacent background, yielding more coherent masks across scales. These visual trends match the quantitative advantage reported next.

As shown in Table 6, D2FLS-Net yields the highest mIoU at 72.86% together with strong Recall at 82.55% and high Precision at 93.30%. The absolute gain in mIoU over SegFormer and SAM is+7.06% and +6.96% respectively, and the gains over UNet, HRNet, and ResNet50 are+11.98%,+8.43%, and +8.99%. While HRNet attains the highest precision at 94.10%, it does so at the cost of substantially lower mIoU at 64.43% and Recall at 70.28%, which suggests missed positives. In contrast, D2FLS-Net delivers a more balanced operating point—higher recall without sacrificing precision—consistent with the intended synergy between DSDF and T-PACE.

Table 7 lists end-to-end inference time on the full Bijie corpus and the full Landslide4Sense2022 corpus, evaluated on an RTX 3090 under a unified protocol (FP16/AMP, batch size=1, forward-only). On Bijie, the fastest methods are ResNet50 (52.3 s) and UNet (66.6 s), followed by SAM (ViT-B, 80.4 s) and HRNet (115.1 s). Our D2FLS-Net finishes in 163.6 s, while SegFormer-B2 is the slowest at 485.3 s. On Landslide4Sense 2022, all methods are quicker in absolute time; ResNet50 (14.5 s) and UNet (19.4 s) remain fastest, SAM (24.2 s) and HRNet (33.9 s) are mid-range, D2FLS-Net requires 43.6 s, and SegFormer-B2 is slowest at 826.6 s. Overall, D2FLS-Net introduces moderate overhead relative to light CNN baselines but remains practical, and together with its accuracy gains offers a reasonable efficiency under the standardized setup.

**3.3.4. Cross-validation.** To evaluate robustness beyond a single split, we performed five-fold stratified cross-validation on both benchmarks, preventing leakage of tiles from the same large scene between train and validation within a fold. All folds share the same optimization and augmentation settings. We aggregate receiver operating characteristic (ROC) curves across folds (mean curve) and display ±1 standard deviation (SD) bands; mean AUC±SD is reported for each method, see Fig 7a for Bijie and Fig 7b for Landslide4Sense 2022. On Bijie it achieves a mean AUC of 0.945±0.008, and on Landslide4Sense 2022 it reaches 0.913±0.012, both with comparatively narrow confidence bands. The relative ordering of methods is preserved from Bijie (0.8 m RGB with 2 m DEM) to Landslide4Sense 2022 (multisensor, 10 m), which suggests that the learned inductive bias is transferable across resolutions and sensors rather than tied to a particular split. Taken together, the cross-validation results support that D2FLS-Net provides accurate, stable, and generalizable performance for automatic landslide segmentation.

## 4. Discussion

The D2FLS-Net constructed in this study shows superior performance in mIoU, precision, and recall compared with other methods on the Bijie and Landslide4Sense2022 dataset. Under five-fold cross-validation, the gains remain, and the stage placement ablation supports the design choice of shallow fusion at Swin-1 and deep fusion at Swin- 4. The experimental

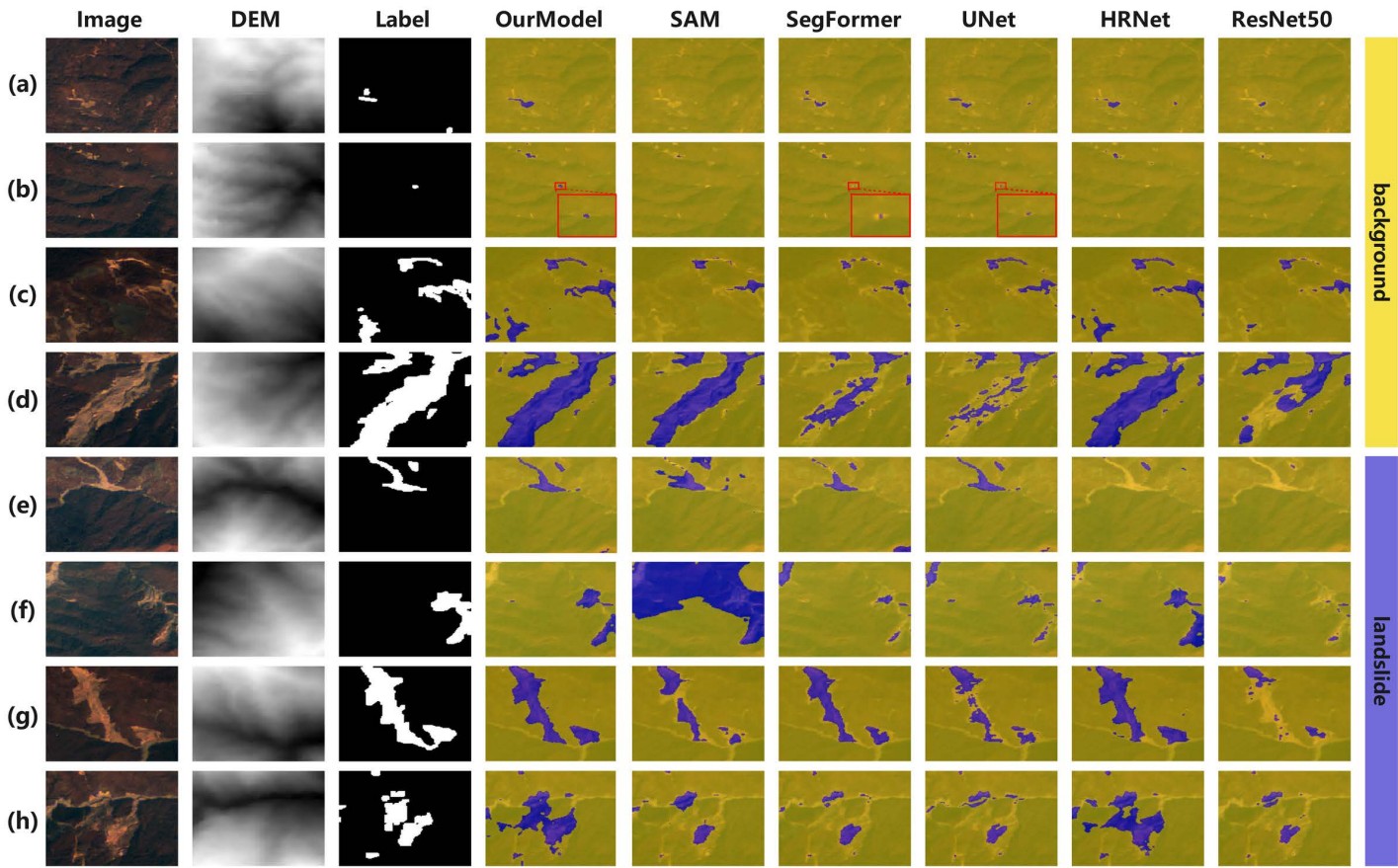

**Fig 6. Visual comparison on the Landslide4Sense2022 dataset. blue=landslide, yellow=backgroun (CC BY 4.0, https://doi.org/10.5281/zenodo.10463239).**

**Table 6. Quantitative comparison on the Landslide4Sense2022 dataset.**

| Method | MIoU(%) | Recall(%) | Precision(%) |
|---|---|---|---|
| D2FLS-Net | 72.86 | 82.55 | 93.30 |
| SAM (ViT-B) | 65.90 | 69.10 | 85.29 |
| SegFormer-B2 | 65.80 | 75.99 | 88.28 |
| UNet | 60.88 | 66.33 | 84.74 |
| HRNet | 64.43 | 70.28 | 94.10 |
| ResNet50 | 63.87 | 72.50 | 87.34 |

results are consistent with our design intent of taking landslide geomorphic characteristics into account. Compared with other methods specifically designed for fusing digital elevation model data and optical data, our approach maintains competitive mIoU. Wang et al. [37] evaluated multiple RGB+DEM architectures, their best configuration, namely a Swin-Transformer with boundary loss, achieved a recall of 89.5% on the Bijie test set, whereas under the same task setting our D2FLS-Net reached a recall of 95.27%. On the Landslide4Sense2022 dataset, the official benchmark reports precision and recall for CNN baseline models trained on stacked multi-source data [38]. Among them, FRRN-B attained the highest recall at 76.16%. D2FLS-Net substantially surpasses this, increasing recall by 6.39% while maintaining

**Table 7. The time complexity.**

| Method | Time(s) on Bijie dataset | Time(s) on Landslide4Sense 2022 dataset |
| --- | --- | --- |
| D2FLS-Net | 163.6 | 43.6 |
| SAM (ViT-B) | 80.4 | 24.2 |
| SegFormer-B2 | 485.3 | 826.6 |
| UNet | 66.6 | 19.4 |
| HRNet | 115.1 | 33.9 |
| ResNet50 | 52.3 | 14.5 |

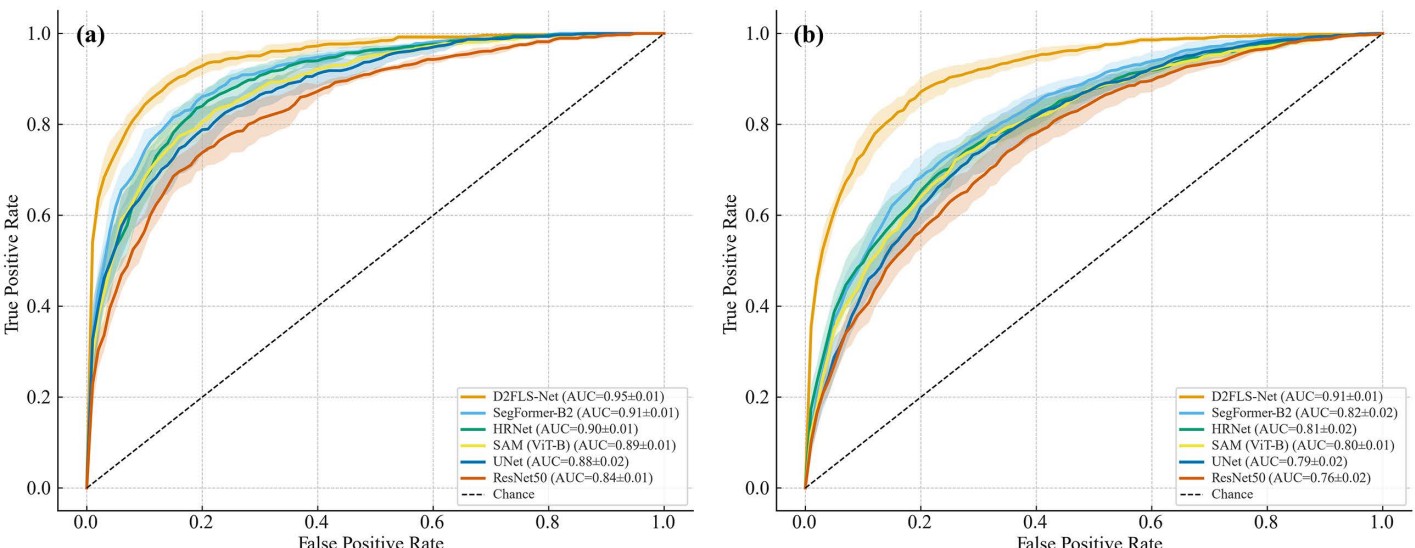

**Fig 7. Receiver Operating Characteristic curve with five-fold cross-validation.** (a) Bijie dataset; (b) Landslide4Sense 2022. Shaded bands denote 95% confidence intervals across folds; the dashed diagonal indicates chance. Legends report mean AUC ± s.d. over the five folds.

competitive mIoU. In disaster mitigation, priority is to reduce missed reports (false negatives), followed by limiting false reports (false positives) in landslide mapping [39]. Therefore, a higher recall is more desirable. These results indicate that, relative to other data-fusion methods, D2FLS-Net is more advantageous. The present work has several limitations. Although the method attains strong accuracy related metrics, its interpretability may be challenging. It also requires more globally diverse landslide data for training, and the model should be appropriately lightened to deployment in automated landslide recognition systems.

## 5. Conclusion

Landslide recognition is a challenging task. To verify the effectiveness of fusing optical data with digital elevation model data in D2FLS-Net, we conducted studies on the Bijie dataset and the Landslide4Sense2022 dataset, and the following conclusions can be drawn.

(1) This study constructs the DSDF module to inject terrain features in stages in a persistent manner and to enable inter-action with optical features, so as to meet the recognition needs of landslides with different morphologies. The T PACE module is constructed to balance the background that must be attended to for landslide recognition and the require-ment for fine segmentation.

(2) The experimental results further show that after averaging over three backbone networks of different capacity, and compared with a model that uses only the basic Swin module, the full model with DSDF and T PACE improves mean Intersection over Union, recall, and precision on the Bijie dataset by 13.6%, 14.2%, and 1.0%. On the Landslide4Sense2022 dataset, the corresponding improvements are 5.5%, 6.1%, and 12.6%.

(3) Compared with five other models including SAM, SegFormer, U Net, HRNet, and ResNet 50, on the Bijie dataset D2FLS-Net attains mean Intersection over Union, recall, and precision that exceed those of the strongest competing model SegFormer by 7.96%, 7.90%, and 4.05%. On the Landslide4Sense2022 dataset, the values exceed those of SegFormer by 7.06%, 6.56%, and 5.02%. This is because D2FLS-Net effectively fuses digital elevation model data and optical data, extracts terrain features, ensures complete extraction of landslides with blurred boundaries, and dynamically adjusts the field of view to achieve finer segmentation, thereby reducing missed detections and false detections.

## Author contributions

**Formal analysis:** Cheng Xiao.

**Funding acquisition:** Chengwei Zhao, Yubin Song.

**Methodology:** Chengwei Zhao.

**Project administration:** Long Li.

**Supervision:** Xuqing Li, Dongsheng Ren.

**Validation:** Chengwei Zhao, Yubo Wang.

**Visualization:** Chengwei Zhao.

**Writing – original draft:** Chengwei Zhao.

**Writing – review & editing:** Chengwei Zhao, Chong Xu.

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
