## [Decision Letter · Decision Letter 0]

1 Sep 2025

Dear Dr. Li,

Thank you for submitting your manuscript to PLOS ONE. After careful consideration, we feel that it has merit but does not fully meet PLOS ONE’s publication criteria as it currently stands. Therefore, we invite you to submit a revised version of the manuscript that addresses the points raised during the review process.

**Editor’s Specific Comments**

In addition to the above reviewer points, I emphasize:

Strengthen the positioning of your contribution. Highlight *what is fundamentally new* compared to existing DEM+optical fusion models.

Discuss how the model could be scaled or deployed in practice (e.g., pilot projects, global applicability).

More detail is needed on why certain datasets/years/resolutions were chosen and how results generalize.

Current quality is not acceptable for publication; must be redrawn.

Needs a deeper scientific comparison, not only performance summary.

We look forward to receiving your revised manuscript.

Kind regards,

Subrata Mondal, M.Phil, Ph.D

Academic Editor

PLOS ONE

Journal Requirements:

5. Please amend either the title on the online submission form (via Edit Submission) or the title in the manuscript so that they are identical.

6. We note that Figures 4-6 in your submission contain [map/satellite] images which may be copyrighted. All PLOS content is published under the Creative Commons Attribution License (CC BY 4.0), which means that the manuscript, images, and Supporting Information files will be freely available online, and any third party is permitted to access, download, copy, distribute, and use these materials in any way, even commercially, with proper attribution. For these reasons, we cannot publish previously copyrighted maps or satellite images created using proprietary data, such as Google software (Google Maps, Street View, and Earth). For more information, see our copyright guidelines: http://journals.plos.org/plosone/s/licenses-and-copyright.

1. You may seek permission from the original copyright holder of Figures 4-6 to publish the content specifically under the CC BY 4.0 license. 

7. We note you have included a table to which you do not refer in the text of your manuscript. Please ensure that you refer to Table 1 in your text; if accepted, production will need this reference to link the reader to the Table.

**Additional Editor Comments:**

Reviewer #1:

This study presents a novel Swin Transformer-based framework (D2FLS-Net) for landslide segmentation, featuring two innovative modules: Dual-Stage DEM-Guided Fusion (DSDF) and Terrain-aware Pixel-wise Adaptive Context Enhancement (T-PACE). The work demonstrates strong technical merits and achieves state-of-the-art performance on two benchmark datasets. The topic would be interesting to readers of PLOS ONE. While the study is partly interesting and valuable, There is a large room for improvement in the language, novelty, contents, and figures of the manuscript. I recommends major revision of this manuscript for the reasons outlined below:

Abstract

(1) Lines 27: module should be deleted

(2) Please add quantitative comparison to DEM fusion baselines in the Abstract, which can enhance the novelty of the manuscript.

(3) The main novelty of the manuscript should be highlighted. Why do authors propose a novel to identify the boundary of landslides? The reasons for developing this study should be clarited.

(4) “remote sensing imaery” and “feature fusion” should be included in Keywords.

Introduction/Methodology

(1) Lines 64-71,114-125. The authors should summary the main achievement of existing literature not simply listing study contents.

(2) Main contribution of the study should be described in Conclusion section, not the end of Introdution section.

(3) Reorganize logic of Introduction section. The author should tell readers why developing this study.

(4) Line 313 F1 equation is not found, and the F1 indicator is also not used to evaluate the performance of the model.

Results and analysis

(1) The manuscript should elaborate on why two specific fusion stages (Swin-1 and Swin-4) were selected rather than intermediate stages. Provide ablation results testing alternative stage combinations.

(2) 700 samples are too few, which can not fully train deep learning models. From Table 1, the number of weighted coefficient (w), and bias (b) are over 1000. The authors should develop cross-validation strategy to illustrate the model robustness.

Conclusions:

(1) This section should tell readers what main contributions of the manuscript is.

Figures

(1) The quality of the figures is too poor to recognize texts.

Reviewer #2:

hello,

The following comments are provided regarding the present paper with the aim of enhancing its quality and quantity. I hope that by applying these points, we will witness an improvement in the reserch’s quality."

- this research was conducted in Bijine City and is a case study. Why isn’t this mentioned in the title?

- in the abstract, you proposed the “Dual-Stage DEM-guided…” method before presenting the results. It’s possible that after reviewing the results, you might decide not to propose this method. Please either correct this or remove it from the abstract section.

- using the acronym “D2FLS-Net” in the title without sufficient explanation might be ambiguous for some readers. It’s better to use the full equivalent of this term in the title, at least.

- in line 6 of the abstract, the word “module” has a typo that needs to be corrected.

- in abstract: Given the claim of “Robust” in the title, the abstract should have elaborated more on how this robustness is achieved (e.g., does the model perform consistently under various data conditions, vegetation cover, or landslide types?). Although the mention of improved Recall in hazy and vegetated areas hints at it, it could have been highlighted more prominently.

- In line 3 of the abstract: Does this imply that previous models didn’t benefit from DEM in some scenarios, and this model always does? More clarity is needed.

- In the keywords section: You could also use “DSDF,” “T-PACE,” “Fusion,” and “DEM-Guided” to increase the searchability of the article.

- Various types of landslides exist, and their causal factors differ in some cases. This point must be taken into consideration.

- L. 52: is it truly 100% feasible in reality to use highly accurate automated systems for identifying these features?

- L. 62: after introducing the challenges and mentioning CNNs, there is a need for a clearer paragraph explaining why existing CNN methods (or even general transformers) are insufficient for these challenges and why the proposed approach (D2FLS-Net) is necessary. This knowledge gap should be articulated more explicitly to provide a stronger justification for the innovation.

- L 64: the reference to “Yang et al.” without sufficient explanation of their achievement is a bit abrupt. It would have been better to briefly explain what their work was and what limitations it had that this paper intends to address.

- in the introduction: the introduction of the article should include a section on the general overview of the research and another section on previous research. These sections should be clearly distinguishable from each other. In this article, this is not observed, and in some sections (e.g., L. 90), the reader becomes confused about which part of the introduction they are in.

- L. 125: Although the abstract referred to “inconsistent benefits of combining DEM with optical data,” the introduction has not yet addressed this issue and its reasons (e.g., why previous models struggled to combine these data). This topic requires fundamental justification.

- L. 137: although the DSDF and T-PACE modules seem innovative, it must be clearly shown how these modules not only improve performance but also specifically solve particular challenges (such as boundary ambiguity and inconsistent DEM benefits). This requires providing a deeper analysis.

- given the claim of “Robust” in the title, the paper should include more analyses demonstrating how the model performs against variations in data quality, landslide types, and different environmental conditions (e.g., in different seasons or with varying vegetation cover). Have experiments been conducted on noisy or diverse data?

- L. 148: the details of the “baselines” and the rationale for their selection should be carefully explained. Does the comparison include methods specifically focused on combining DEM and optical data?

- L. 148: although transformer models are powerful, their interpretability can be challenging. Do the authors have a plan to show how the model uses DEM and RGB information (e.g., with saliency maps or other interpretability techniques)?

+ how does D2FLS-Net address these problems with “dual-stage topographic injection,” “cross-attention,” and “context enhancement”? These should be introduced as direct solutions to the problems raised in the introduction.

+ the abstract mentions “inconsistent benefits of fusing DEMs,” but the introduction has not yet fully clarified this problem for the reader. It should be explained why simply “adding DEM” to CNNs or transformers does not always yield optimal results. Is this due to incorrect alignment of visual and topographic features? Or due to differences in the nature of their information?

+ why are CNNs, despite their powerful capabilities, still insufficient, paving the way for the Transformer?

- L. 171: Why only these resolutions? Is there a specific reason?

- L. 176: Typo!

- from lines 178 to 199: None of the equations are referred to in the text of the article, nor do they have any references.

- L. 200: Typo!

- L. 256: Generally, in areas with vegetation cover, especially forest cover, landslides occur much less frequently. In which part of the article did you consider this?

- from lines 263 to 283: None of the equations are referred to in the text of the article, nor do they have any references.

- L. 293: Why these specific months and year? Given that there’s about a 7-year time gap to the present, is it logical to use images from 2018?

- from lines 308 to 313: None of the equations are referred to in the text of the article, nor do they have any references.

- L. 306: FN, TN, FP, TP must be clearly defined as to what factors they represent and how they are calculated.

- L. 326: the results section should preferably start with a general introduction and text rather than a table. This point must be taken into consideration.

- in the discussion and conclusion sections: this section is written very generally and briefly, lacking any relevant discussion and merely reviewing the results from the previous section. Furthermore, in this section, you have not compared your research findings with those of other researchers. This section definitely needs to be rewritten. Also, please provide some relevant suggestions related to your research, especially concerning the study area, that would lead to solving a problem (please pay special attention to this point).

- page 36: The quality of the figures is very low.

- page 39: No cartographically standard map of the study area has been provided. Unfortunately.

- page 40: It appears that SAM identifies areas with more severe landslides much more accurately!

+ were any field visits conducted to the landslides extracted from the images to verify the accuracy of the image-derived results?

+ none of the figures have legends! Unfortunately!

+ the current paper is theoretically very strong but has fundamental weaknesses in its practical and implementable aspects.

+ the paper does not have a good literature review and has not adequately answered the question of why this research was conducted.

+ based on the article, it can be predicted that the paper was prepared quickly, and not much time was spent on improving its quality and quantity.

+ what are the CELL SIZES of the DEMs you used?

+ does the time of landslide occurrence also affect your research results? Meaning, for example, a landslide that occurred in 2014 versus one in 2018. How can the impact of this 4-year difference be eliminated so as not to affect the results?

+ can the volume of material displaced by the landslide and the area of the landslide also affect the results?

+ does the orientation of the hillslopes also affect the results?

+ does the general shape of the hillslopes also affect the results?

+ does the actual shape of the hillslopes (e.g., convexity, concavity, divergence, convergence, etc.) also affect the research results?

+ to improve the logical flow, the introduction should include a clear “Our Contributions” section that summarizes the innovations of D2FLS-Net in bullet points, with reference to the challenges raised. This helps the reader know exactly what to expect from the rest of the paper.

+ how confident are you in the results of the current research such that if you were asked to implement it as a pilot project in one area, and if successful, to implement the results across an entire province, country, or even globally; would you agree to this proposal?

Reviewers' comments:

Reviewer's Responses to Questions

**Comments to the Author**

1. Is the manuscript technically sound, and do the data support the conclusions?

Reviewer #1: Yes

Reviewer #2: Partly

2. Has the statistical analysis been performed appropriately and rigorously?

Reviewer #1: No

Reviewer #2: N/A

3. Have the authors made all data underlying the findings in their manuscript fully available?

Reviewer #1: No

Reviewer #2: No

4. Is the manuscript presented in an intelligible fashion and written in standard English?

Reviewer #1: No

Reviewer #2: Yes

Reviewer #1: Abstract

(1) Lines 27: module should be deleted

(2) Please add quantitative comparison to DEM fusion baselines in the Abstract, which can enhance the novelty of the manuscript.

(3) The main novelty of the manuscript should be highlighted. Why do authors propose a novel to identify the boundary of landslides? The reasons for developing this study should be clarited.

(4) “remote sensing imaery” and “feature fusion” should be included in Keywords.

Introduction/Methodology

(1) Lines 64-71,114-125. The authors should summary the main achievement of existing literature not simply listing study contents.

(2) Main contribution of the study should be described in Conclusion section, not the end of Introdution section.

(3) Reorganize logic of Introduction section. The author should tell readers why developing this study.

(4) Line 313 F1 equation is not found, and the F1 indicator is also not used to evaluate the performance of the model.

Results and analysis

(1) The manuscript should elaborate on why two specific fusion stages (Swin-1 and Swin-4) were selected rather than intermediate stages. Provide ablation results testing alternative stage combinations.

(2) 700 samples are too few, which can not fully train deep learning models. From Table 1, the number of weighted coefficient (w), and bias (b) are over 1000. The authors should develop cross-validation strategy to illustrate the model robustness.

Conclusions:

(1) This section should tell readers what main contributions of the manuscript is.

Figures

(1) The quality of the figures is too poor to recognize texts.

Reviewer #2: hello,

The following comments are provided regarding the present paper with the aim of enhancing its quality and quantity. I hope that by applying these points, we will witness an improvement in the reserch’s quality."

- this research was conducted in Bijine City and is a case study. Why isn’t this mentioned in the title?

- in the abstract, you proposed the “Dual-Stage DEM-guided…” method before presenting the results. It’s possible that after reviewing the results, you might decide not to propose this method. Please either correct this or remove it from the abstract section.

- using the acronym “D2FLS-Net” in the title without sufficient explanation might be ambiguous for some readers. It’s better to use the full equivalent of this term in the title, at least.

- in line 6 of the abstract, the word “module” has a typo that needs to be corrected.

- in abstract: Given the claim of “Robust” in the title, the abstract should have elaborated more on how this robustness is achieved (e.g., does the model perform consistently under various data conditions, vegetation cover, or landslide types?). Although the mention of improved Recall in hazy and vegetated areas hints at it, it could have been highlighted more prominently.

- In line 3 of the abstract: Does this imply that previous models didn’t benefit from DEM in some scenarios, and this model always does? More clarity is needed.

- In the keywords section: You could also use “DSDF,” “T-PACE,” “Fusion,” and “DEM-Guided” to increase the searchability of the article.

- Various types of landslides exist, and their causal factors differ in some cases. This point must be taken into consideration.

- L. 52: is it truly 100% feasible in reality to use highly accurate automated systems for identifying these features?

- L. 62: after introducing the challenges and mentioning CNNs, there is a need for a clearer paragraph explaining why existing CNN methods (or even general transformers) are insufficient for these challenges and why the proposed approach (D2FLS-Net) is necessary. This knowledge gap should be articulated more explicitly to provide a stronger justification for the innovation.

- L 64: the reference to “Yang et al.” without sufficient explanation of their achievement is a bit abrupt. It would have been better to briefly explain what their work was and what limitations it had that this paper intends to address.

- in the introduction: the introduction of the article should include a section on the general overview of the research and another section on previous research. These sections should be clearly distinguishable from each other. In this article, this is not observed, and in some sections (e.g., L. 90), the reader becomes confused about which part of the introduction they are in.

- L. 125: Although the abstract referred to “inconsistent benefits of combining DEM with optical data,” the introduction has not yet addressed this issue and its reasons (e.g., why previous models struggled to combine these data). This topic requires fundamental justification.

- L. 137: although the DSDF and T-PACE modules seem innovative, it must be clearly shown how these modules not only improve performance but also specifically solve particular challenges (such as boundary ambiguity and inconsistent DEM benefits). This requires providing a deeper analysis.

- given the claim of “Robust” in the title, the paper should include more analyses demonstrating how the model performs against variations in data quality, landslide types, and different environmental conditions (e.g., in different seasons or with varying vegetation cover). Have experiments been conducted on noisy or diverse data?

- L. 148: the details of the “baselines” and the rationale for their selection should be carefully explained. Does the comparison include methods specifically focused on combining DEM and optical data?

- L. 148: although transformer models are powerful, their interpretability can be challenging. Do the authors have a plan to show how the model uses DEM and RGB information (e.g., with saliency maps or other interpretability techniques)?

+ how does D2FLS-Net address these problems with “dual-stage topographic injection,” “cross-attention,” and “context enhancement”? These should be introduced as direct solutions to the problems raised in the introduction.

+ the abstract mentions “inconsistent benefits of fusing DEMs,” but the introduction has not yet fully clarified this problem for the reader. It should be explained why simply “adding DEM” to CNNs or transformers does not always yield optimal results. Is this due to incorrect alignment of visual and topographic features? Or due to differences in the nature of their information?

+ why are CNNs, despite their powerful capabilities, still insufficient, paving the way for the Transformer?

- L. 171: Why only these resolutions? Is there a specific reason?

- L. 176: Typo!

- from lines 178 to 199: None of the equations are referred to in the text of the article, nor do they have any references.

- L. 200: Typo!

- L. 256: Generally, in areas with vegetation cover, especially forest cover, landslides occur much less frequently. In which part of the article did you consider this?

- from lines 263 to 283: None of the equations are referred to in the text of the article, nor do they have any references.

- L. 293: Why these specific months and year? Given that there’s about a 7-year time gap to the present, is it logical to use images from 2018?

- from lines 308 to 313: None of the equations are referred to in the text of the article, nor do they have any references.

- L. 306: FN, TN, FP, TP must be clearly defined as to what factors they represent and how they are calculated.

- L. 326: the results section should preferably start with a general introduction and text rather than a table. This point must be taken into consideration.

- in the discussion and conclusion sections: this section is written very generally and briefly, lacking any relevant discussion and merely reviewing the results from the previous section. Furthermore, in this section, you have not compared your research findings with those of other researchers. This section definitely needs to be rewritten. Also, please provide some relevant suggestions related to your research, especially concerning the study area, that would lead to solving a problem (please pay special attention to this point).

- page 36: The quality of the figures is very low.

- page 39: No cartographically standard map of the study area has been provided. Unfortunately.

- page 40: It appears that SAM identifies areas with more severe landslides much more accurately!

+ were any field visits conducted to the landslides extracted from the images to verify the accuracy of the image-derived results?

+ none of the figures have legends! Unfortunately!

+ the current paper is theoretically very strong but has fundamental weaknesses in its practical and implementable aspects.

+ the paper does not have a good literature review and has not adequately answered the question of why this research was conducted.

+ based on the article, it can be predicted that the paper was prepared quickly, and not much time was spent on improving its quality and quantity.

+ what are the CELL SIZES of the DEMs you used?

+ does the time of landslide occurrence also affect your research results? Meaning, for example, a landslide that occurred in 2014 versus one in 2018. How can the impact of this 4-year difference be eliminated so as not to affect the results?

+ can the volume of material displaced by the landslide and the area of the landslide also affect the results?

+ does the orientation of the hillslopes also affect the results?

+ does the general shape of the hillslopes also affect the results?

+ does the actual shape of the hillslopes (e.g., convexity, concavity, divergence, convergence, etc.) also affect the research results?

+ to improve the logical flow, the introduction should include a clear “Our Contributions” section that summarizes the innovations of D2FLS-Net in bullet points, with reference to the challenges raised. This helps the reader know exactly what to expect from the rest of the paper.

+ how confident are you in the results of the current research such that if you were asked to implement it as a pilot project in one area, and if successful, to implement the results across an entire province, country, or even globally; would you agree to this proposal?

**Do you want your identity to be public for this peer review?** For information about this choice, including consent withdrawal, please see our Privacy Policy

Reviewer #1: No

Reviewer #2: No

---

## [Author Response · Author response to Decision Letter 1]

5 Oct 2025

We are pleased to have the privilege of receiving the reviewers’ comments. We thank the reviewers for considering our manuscript and for providing comments and the opportunity to revise. We have made revisions and added experiments in accordance with the editors’ and reviewers’ comments. Please see the response to reviewers.docx for details.

---

## [Editor Report · Decision Letter 1]

16 Oct 2025

Dear Dr. Li,

Thank you for submitting your manuscript to PLOS ONE. After careful consideration, we feel that it has merit but does not fully meet PLOS ONE’s publication criteria as it currently stands. Therefore, we invite you to submit a revised version of the manuscript that addresses the points raised during the review process.

**ACADEMIC EDITOR:**

The manuscript is suitable for publication, but a few minor adjustments to sentence structure, explanations, and figure clarity will enhance its readability and impact. If the authors make these adjustments, the manuscript will be ready for publication.

Some sentences, especially in the Methodology and Discussion sections, can be simplified or broken into smaller parts for better readability and flow.

A few technical terms like "multi-dilation atrous branches" and "boundary localization" could benefit from more intuitive explanations or additional context for readers less familiar with the specifics.

Ensure that all figures and tables are referenced properly and are clearly labeled, which will help readers interpret the results more easily

We look forward to receiving your revised manuscript.

Kind regards,

Subrata Mondal, M.Phil, Ph.D

Academic Editor

PLOS ONE
---

## [Author Response · Author response to Decision Letter 2]

21 Oct 2025

We are pleased and grateful for the opportunity to submit our work to your esteemed journal and for the recognition it has received from the Editor and reviewers. We also thank the Editor for the time and effort devoted to the review of our manuscript. In response to the most recent comments, we provide the following point-by-point replies and have made targeted revisions accordingly.

1. Some sentences, especially in the Methodology and Discussion sections, can be simplified or broken into smaller parts for better readability and flow.

[Response]

We appreciate your valuable suggestions, which have clearly pointed out areas for improvement in our paper.

[Changes made]

In response to the Editor’s comment, we have revised the manuscript throughout by splitting overly long and complex sentences to improve clarity and flow, particularly in the Methodology and Discussion sections. We also replaced several informal expressions with standard academic wording to enhance readability.

2. A few technical terms like "multi-dilation atrous branches" and "boundary localization" could benefit from more intuitive explanations or additional context for readers less familiar with the specifics.

[Response]

Thank you for your comments, which provide constructive guidance on improving the manuscript’s readability and will help the paper better serve a diverse readership.

[Changes made]

After the two terms noted by the Editor, we added:

“parallel dilated convolution branches with different dilation rates (multi-dilation atrous branches)” (see line 138).

“Together, DSDF and T-PACE enhance boundary localization—i.e., precise edge delineation for vegetated and ancient landslides—while preserving robustness for new landslides.” (see line 140).

Following the cross-attention (MHCA) module, we added:

“Cross-attention here can be read as letting RGB ‘ask questions’ and the DEM ‘provide context’, so high-level RGB features retrieve terrain cues that are consistent with large-scale geomorphology.” (see lines 234–236).

We further clarified the FPN description as:

“a lightweight Feature Pyramid Network (FPN) decoder head, a standard module for multi-scale feature fusion, to generate the final two-channel mask.” (see line 147).

We also explained the term “logits” as:

“logits (the raw scores before converting to class probabilities)” (see line 181).

3. Ensure that all figures and tables are referenced properly and are clearly labeled, which will help readers interpret the results more easily

[Response]

We thank the Academic Editor for the helpful comment. Upon re-examining the manuscript, we confirmed that there are instances of duplicated citations.

[Changes made]

In response to the comments, we corrected minor errors in several figures. We also verified that all Figures and Tables are cited in numerical order and in accordance with journal style; specifically, we adjusted their placement so that each item appears in the main text immediately after its first in-text citation. The figures were prepared in Visio and exported at the highest resolution, then converted using PACE as required by PLOS to ensure compliance with the journal’s figure specifications. In addition, we added necessary explanatory details to the figure captions.

---

## [Editor Report · Decision Letter 2]

10 Nov 2025

D2FLS-Net:Dual-Stage DEM-guided Fusion Transformer for landslide segmentation

PONE-D-25-44116R2

Dear Dr. Li,

We’re pleased to inform you that your manuscript has been judged scientifically suitable for publication and will be formally accepted for publication once it meets all outstanding technical requirements.

Kind regards,

Subrata Mondal, M.Phil, Ph.D

Academic Editor

PLOS ONE
---

## [Editor Report · Acceptance letter]

PONE-D-25-44116R2

PLOS ONE

Dear Dr. Li,

I'm pleased to inform you that your manuscript has been deemed suitable for publication in PLOS ONE. Congratulations! Your manuscript is now being handed over to our production team.

Kind regards,

on behalf of

Dr. Subrata Mondal

Academic Editor

PLOS ONE